# The Use of Dansyl Chloride to Probe Protein Structure and Dynamics

**DOI:** 10.3390/ijms26020456

**Published:** 2025-01-08

**Authors:** James Larson, Monika Tokmina-Lukaszewska, Jadyn Malone, Ethan J. Hasenoehrl, Will Kelly, Xuelan Fang, Aidan White, Angela Patterson, Brian Bothner

**Affiliations:** 1Department of Chemistry and Biochemistry, Montana State University, Bozeman, MT 59717, USA; 2Department of Molecular and Cellular Biochemistry, Indiana University, Bloomington, IN 47405, USA

**Keywords:** dansyl chloride, protein chemistry, protein labeling, protein structure, protein dynamics, native mass spectrometry, ion mobility, mass spectrometry

## Abstract

Dansyl labeling is a widely used approach for enhancing the detection of small molecules by UV spectroscopy and mass spectrometry. It has been successfully applied to identify and quantify a variety of biological and environmental specimens. Despite clear advantages, the dansylation reaction has found very few applications in the study of proteins. We reasoned that the mild labeling conditions, small size, and rapid reaction could be beneficial for studying protein structure and dynamics. To test this, we investigated the impact of dansylation on protein fold, stability, protein–protein, and protein–cofactor interactions. We selected two model proteins, myoglobin and alcohol dehydrogenase, for analysis using native mass spectrometry and ion mobility mass spectrometry. Our work establishes the utility of dansyl chloride as a covalent probe to study protein structure and dynamics under native conditions.

## 1. Introduction

Proteins are vital macromolecules that achieve function through their structure and dynamics [1,2]. Labeling of proteins through chemical modifications using molecular probes emerged as a technique to understand the tertiary and quaternary structures of proteins, protein–protein, and protein–ligand interactions, and how proteins and protein complexes change under different conditions (presence of a ligand, pH, temperature, etc.) [3,4]. Results from these experiments allow for a fundamental knowledge of protein structure and dynamics, which ultimately facilitates a deeper understanding of protein function.

Soluble proteins adopt globular forms with charged amino acids such as lysine and glutamate being exposed on the surface, accessible to solvents [5]. Lysine is a primary target for protein chemists due to its relative abundance and nucleophilicity [6]. Common lysine-specific commercial probes include acetic anhydride, succinic anhydride, N-hydroxysuccinimide (NHS) ester derivatives, and isothiocyanates derivates [3,6]. The use of these probes is a powerful tool in the investigation of protein structure and function. For example, succinic anhydride has been used to help elucidate the structure of general diffusion porins from *Rhodobacter capsulatus* in which three lysine residues line the pore channel providing new insights into the structure and conductance of general porins [7]. Additionally, the isothiocyanate derivate, fluorescein 5′-isothiocyante, was used to locate the binding site of human lactotransferrin using fluorescein 5′-isothiocyanate providing insights into the mechanism of iron delivery to enterocytes and peripheral blood lymphocytes [8]. NHS and isothiocyanate derivates are generally tethered to large fluorophores (like fluorescein 5′-isothiocyanate), which additionally allow for fluorescence resonance energy transfer (FRET) and fluorescence microscopy studies [9,10,11]. Despite the widespread use of labeling of amino acid residues in structural protein studies, these techniques can introduce artifacts because chemical modification can alter local structure, hydrophobicity, and function [12,13]. Indeed, labeling of cysteine residues of the viral capsid proteins of *Nudaueliai ω capensis* with a naphthalene-conjugated maleimide dye prevented conformational changes required for maturation [14]. A review of mass spectrometry studies using amino acid-specific labeling found that over 60% of studies did not assess the structural integrity of the protein of interest post-labeling [3]. We see this as a significant gap in understanding the use and impact of protein labeling.

We propose that dansyl chloride (DnsCl) is an excellent candidate for covalently labeling solvent-accessible lysine residues. DnsCl covalently reacts preferentially and rapidly with primary amines (35 M^−1^s^−1^) [15,16] and has been extensively utilized to label the amine-containing small molecules [17,18], amino acids, and the N terminus of peptides for sequencing [19,20,21,22]. Despite its historical uses, there are few reports of the application of DnsCl labeling for structural studies of native proteins or protein complexes [23,24,25]. As a probe, DnsCl is appealing because it offers several advantages over other commercially available labels. While DnsCl is similar in size to NHS and succinic anhydride, it is endowed with fluorescence and UV-vis properties. Reactive groups such as NHS, succinic anhydride, and isothiocyanate require the addition of bulky fluorophores to be spectroscopically active (Figure 1) [26,27]. Moreover, the dansyl (Dns) moiety is proven to increase chromatographic separation and enhance ionization in liquid chromatography–mass spectrometry (LC–MS) analysis, therefore making this protocol ideal for any bottom-up investigation. Due to the presence of the UV-active mass tag, the data processing time is much shorter than that of the classic “footprinting” approach. Even though DnsCl solubility in water is rather low, it is actually an advantage while conducting structural studies. The naturally low probe concentration in aqueous solution helps prevent protein structural changes induced by over-labeling and makes a sample cleanup optional. DnsCl is reactive in a wide pH and temperature range, offering undeniable protocol flexibility, which is especially important for labeling challenging protein systems. The final advantage is its cost; DnsCl can be 1/4–1/200 the cost on a weight-to-weight basis (Figure 1). Together, these properties advocate for investigating DnsCl as a chemical probe for the structural analysis of proteins and protein complexes. In our previous work, we demonstrated the successful use of DnsCl to investigate protein–protein interactions. We used that information alongside other, non-mass spectrometry-derived outcomes to generate protein complex models of flavodiiron proteins (Flv1 and Flv3) from cyanobacteria *Synechocystis* sp. PCC 6803 and electron transfer flavoprotin (EtfABCX) from hyperthermophilic archaeon *Pyrobaculum aerophilum* [23,24]. Here, we went beyond a “static picture” and presented how the impact of the molecular probe attachments on structural changes and protein dynamics should be investigated.

In this work, we explore the efficacy of dansyl chloride as a probe for the investigation of protein structure and dynamics in their native-like form. To achieve this, we first characterized the stability of dansyl chloride in aqueous environments with UV-vis spectroscopy. We then used native mass spectrometry and ion mobility spectrometry to investigate the impact of dansylation on protein structure. Myoglobin and alcohol dehydrogenase were used to model a generic small monomeric protein with a non-covalently bound cofactor and a large multimeric protein, respectively. Labeling efficiency under different conditions was quantified using liquid chromatography–mass spectrometry. This work also demonstrates the power of native mass spectrometry and ion mobility for accessing structural integrity of proteins and complexes.

## 2. Results

### 2.1. Evaluation of Dansyl Chloride Stability in Aqueous Solutions

To label proteins, DnsCl has to be dissolved in an organic solvent and then added to an aqueous solution containing the protein of interest. However, in aqueous environments, DnsCl hydrolyzes forming the dansyl acid, which is UV-Vis active but inactive in labeling conditions. To optimize conditions for protein labeling, we first investigated the chemical stability of dansyl chloride in aqueous conditions. Based on the literature, the optimal conditions for primary amines and dansyl conjugation are non-primary amine buffers at high pH (~9.5), elevated temperature, and various amounts of organic solvent [22,28,29]. Using UV-Vis spectroscopy, we measured the hydrolysis rate of DnsCl in aqueous solution at room temperature across four pH conditions in non-primary amine buffers: 100 mM carbonate-bicarbonate pH 9.5, 100 mM HEPES pH 8.2, and 100 mM sodium phosphate pH 7 and pH 6. In addition, we tested ammonium acetate solutions at pH 7 and pH 9.5. Saturated DnsCl in acetonitrile (MeCN) was added to the aqueous solutions to 20% volume. Figure 2a presents the UV-Vis results for dansyl chloride hydrolysis in phosphate buffer pH 7. The first spectrum recorded immediately after mixing DnsCl (zero minutes) has an absorbance maximum of 365 nm which is characteristic of DnsCl (substrate). Over time, the dansyl acid product appeared with an absorbance maximum of 315 nm. The solution was completely depleted of the reactive form after 50 min in phosphate buffer pH 7 at room temperature. The rate of DnsCl hydrolysis was correlated with pH in non-amine-containing buffer (Figure 2b). As expected, in ammonium acetate, the rate of DnsCl hydrolysis was faster than in the non-amine-containing solutions of the same pH due to the parallel substitution reaction between the buffer amine and DnsCl.

### 2.2. Characterization of Model Protein Systems Stability in Labeling Conditions

Organic solvents alter the polarity of a solution which in turn may cause changes in protein structure and even denaturation. Because the chemical probes must be solubilized in organic solvent prior to protein labeling, we assayed the effect of MeCN on protein structure. For this test, we used myoglobin, a small monomeric protein (17 kDa) with a non-covalently bound heme cofactor, in pH 7 and pH 9.5 ammonium acetate with 0, 2, and 20% MeCN with native mass spectrometry (NMS). In the absence of MeCN, we observed a compact distribution of protein charge states (+6 to +9) with a dominant charge state of +8 for both apo- and holo-myoglobin (Figure 3a). The structure of myoglobin remained unchanged with the addition of 2% and 20% MeCN as well as in the presence/absence of organic solvent at pH 9.5 (Figure 3b, Appendix A). We then challenge holo-myoglobin structure stability by increasing collision energy (25–80 eV) in the trap region. This type of “complex activation” should result in a gradual structure unfolding due to a loss of protein–protein interactions. As the trap energy increased, we observed partial loss of heme at a collision energy of 45 eV. At 60 and 80 eV, most of myoglobin was in the apo-form, and the structure partially collapsed with the dominant charge state shifting to +7. The same trends were observed at pH 9.5. Overall, these data show that at pH 7 and 9.5, with 2 and 20% of MeCN, the structure of myoglobin is unchanged and the heme cofactor remains largely bound.

In our second model protein system, we were interested in the effect organic solvent and pH may have on protein–protein interactions within large multimeric proteins. To investigate this, we performed the same NMS analysis using alcohol dehydrogenase (ADH), a large homotetramer (150 kDa), as we did with myoglobin. At pH 7 and without MeCN, a mixture of monomer, dimer, and tetramer was observed, with the dominant population being the tetramer (Figure 4a). When various amounts of MeCN were added, the tetramer remained the dominant population (Figure 4b,c). To further challenge complex stability, we gradually increased collision energy in a trap region. Within the tested range, the ADH population distribution was unaffected (Figure 4). At pH 9.5, ADH showed a mixed population of monomer, dimer, and tetramer stoichiometries in the presence of 0, 2, and 20% MeCN (Appendix A). Despite partial disassociation of the complex, the subunits remained folded with compact Gaussian charge state distributions. These findings show that at higher pH, the increasing amounts of MeCN have modest effects on the ADH oligomeric state without any major monomer unfolding event.

### 2.3. Robustness of Protein Dansylation Reaction

To determine the extent of labeling following the reaction we implemented an intact protein liquid chromatography–mass spectrometry (LC-MS) approach. The myoglobin samples prepared in the presence of 2% MeCN in biologically relevant buffers (phosphate buffers at pH 6 and 7, and carbonate-bicarbonate buffer at pH 9.5) were exposed to DnsCl for 5, 15, and 30 min and then immediately analyzed with LC/MS. The example of the raw data and deconvoluted mass spectrum of the myoglobin labeled at pH 9.5 for 5 min is presented in Figure 5, panels a and b, respectively. Our results show the presence of multiple species separated by 234 Daltons, indicating successful labeling of the myoglobin species in solution with DnsCl in all tested conditions. Measured *m*/*z* errors for all proteins were less than 0.5 *m*/*z* from calculated values. As expected, labeling at pH 6 resulted in the most modest labeling with 0–2 labels across all time points (Figure 5c). The dominant population was consistently myoglobin with one dansyl at pH 6. At pH 7, the dominant population was also one dansyl, but after 15 min, no population existed with 0 dansyl labels, and a three-dansyl population began to emerge. At pH 9.5, the most robust labeling occurred with a distribution of 1–5 dansyls observed at 5 min. After 15 min, there was less of the 1-dansyl species and more of the 2–5 labeled species. These data show that dansylation occurs over a wide pH range with increased labeling efficiency at pH 9.5.

### 2.4. Dansylation-Induced Structural Changes

Chemical labeling and in particular the attachment of fluorescent tags can alter the native protein structure. In the interest of the application of IMS to investigate protein structural changes and limiting sample handling, we labeled our model systems for 5, 15, and 30 min in 100 mM ammonium acetate pH 7 and 9.5 in the presence of 2% of MeCN (with the exception of myoglobin labeled at pH 6). NMS spectra of myoglobin dansylated for 5 min at pH 7 showed that both apo- and holo-myoglobin species were slightly labeled (maximum 2 Dns label incorporated) and that the charge state distribution was unaffected with the dominant charge state being +8 (Figure 6a). When we challenged the structural stability further with increased trap collisional energy, we saw a shift towards a lower dominant charge state, as we previously observed in the absence of the dansylation reaction. As expected, label incorporation increased with reaction time (Figure 6b,c). Additionally, we did not notice a substantial loss of the heme cofactor with increased labeling time. Next, we turned our attention to measured drift times to detect any conformational changes induced by labeling (Figure 6d,e). Only one conformation was detected for each charge state (Figure 6d) and increasing the number of labels caused subtle changes in the collisional cross-section (CCS) (Figure 6e, Appendix A). Labeling of myoglobin at pH 9.5 yielded similar results with a shift towards charge state +7 with increasing energy and small changes in ion mobility due to label load (Appendix A). As with myoglobin, subtle changes were detected in the ion mobility analysis of ADH between unlabeled and labeled species (Figure 7, Appendix A). Together, these data show that labeling with dansyl chloride does not cause unfolding or significant conformational changes to these mode protein systems.

## 3. Discussion

Conjugation to a nucleophilic side chain of canonical amino acids is the general strategy used by protein chemists for labeling. Lysine is one of the most commonly targeted residues for studying proteins in their native structure because of its nucleophilic amine group and relative abundance [6,30]. In this work, we used two protein systems to develop a robust lysine dansylation protocol which can be used to investigate changes to protein structure and dynamics under native conditions. We used myoglobin as a model for a small, monomeric, cofactor-containing protein, and ADH as a model for a large multimeric protein complex. To distinguish the impact of label incorporation on protein structure from the effects induced by buffer components, the structural stability of model proteins was challenged by changing the pH environment and organic solvent concentration. In addition, we put significant effort into the evaluation of label incorporation on protein structure because there are many published reports where that effect is basically ignored, and conclusions are drawn without any further verification. We used NMS and IMS protocols to demonstrate the suitability of our dansylation protocols for detecting changes in protein conformation.

A significant challenge for protein chemists is balancing reaction conditions that favor high yield and reasonable kinetics while maintaining native protein structure. The small molecule chemistry generally favors organic solvents, high pH, and elevated temperatures. Because these optimal labeling conditions are not suitable for the analysis of protein structure, we investigated the feasibility of the dansylation reaction with more biological relevance from the protein standpoint. Because DnsCl has high reactivity in the nucleophilic substitution, we began our investigation by screening several buffering systems to evaluate a label loss due to the presence of other nucleophiles (other than the lysine amino group) in the protein solution (Figure 2). Based on these results, we concluded that at room temperature, the dansylation reaction could be limited to 30 min and even as short as 10 min at higher pH conditions for a single dose of DnsCl. If slower labeling is required, the reaction can potentially be carried out at lower temperatures. Since DnsCl is introduced into the protein solution in organic solvent, the change in polarity of the working solution should be anticipated and the impact on the protein structure should be evaluated before actual labeling. While we observed no unfolding of our model proteins even at 20% MeCN (Figure 3 and Figure 4), the additive effect of organic solvent and high pH had some impact on the oligomeric state and ratio of apo to holo form. Therefore, we would advise a more cautious approach when higher amounts of organic solvent are required.

In the next step, we investigated the impact of pH on the dansylation reaction. As expected, fewer Dns moieties were incorporated at lower pH and in shorter time (Figure 5). In addition, due to the difference in labeling rates within the pH range 6–9.5, we noticed variability in the labeled species ratios. Therefore, careful pH modulation might be used as an alternative strategy to time course or temperature range to track changes to solvent-exposed surface lysine residues. Many of our labeling experiments were performed in ammonium acetate, which introduces an additional source of the free amine. We chose this solution because labeling in ammonium acetate requires no cleanup step (buffer exchange/dialysis) before NMS/IMS analysis, thus allowing for faster sample preparation and limited sample loss. Despite the side reaction (label loss due to additional nucleophiles present in protein solution), up to three dansyl labels were incorporated onto myoglobin in ammonium acetate at pH 7 (Figure 6). Importantly, dansyl labeling did not interfere with non-covalent protein–ligand interactions as the heme cofactor remained bound to myoglobin, even with three Dns moieties incorporated (Figure 6 and Appendix A). In addition, the calculated CCS values confirmed the lack of any structural change due to DnsCl conjugation to myoglobin in the 6–9.5 pH range, for both the apo and holo form. Similarly, Dns moiety incorporation into ADH had minimal to no effect on protein tertiary structure since we did not observe any significant change in the charge state distribution and small changes to drift time, in addition to that already present before label introduction partial complex dissociation at pH 9.5 (Figure 7 and Appendix A).

Using two model proteins, myoglobin and ADH, we showed that DnsCl is a robust fluorescent probe, allowing for covalent labeling of solvent-accessible lysine residues at a pH range from 6 to 9.5. We showed that the introduction of several dansyl moieties has a negligible effect on non-covalent interactions, nor does it alter the native-like conformation of proteins or protein complexes. We took advantage of collisional-induced unfolding to further improve our sensitivity in detecting small changes in protein structure and dynamics. In addition, we demonstrated that it is feasible to perform dansylation of lysine residues where competitive reactions occur with solution components, such as with ammonium acetate. This allows for the implementation of NMS and IMS protocols to investigate structural change/protein dynamics using dansyl chloride. Labeling of solvent-accessible lysine in conjunction with NMS, and peptide-level MS can be used together to identify protein–protein interactions as unlabeled lysine residues are expected at the interface. Additionally, this combination of techniques can be used to identify ligand-binding sites where dansyl labeling may prevent ligand binding. Conformational change can also be investigated in a protein system where an allosteric effector increases or decreases the number of exposed lysine residues. The dansyl probe is also versatile allowing for selective labeling of different reactive groups, permitting labeling of different amino acid side chains [31,32]. Finally, we provided an experimental framework that should be considered as an evaluation guideline to properly investigate the impact of buffer components/molecular probes when a new protein system is subjected to surface labeling in native conditions.

## 4. Materials and Methods

### 4.1. Chemical Stability of the Dansyl Chloride Probe

The stability of dansyl chloride (DnsCl) in aqueous environments reactivity was measured using UV-Vis spectroscopy on a Nanodrop 2000C (Thermo-Fisher, Waltham, MA, USA) at different pHs, in different solutions and buffers. The solutions and buffers tested were 100 mM carbonate-bicarbonate (Sigma, St. Louis, MO, USA) pH 9.5, 100 mM HEPES (Thermo-Fisher) pH 8.2, 100 mM sodium phosphate (Thermo-Fisher) pH 7, 100 mM sodium phosphate (Thermo-Fisher) pH 6, 100 mM ammonium acetate (J.T. Baker, Phillipsburg, NJ, USA) pH 7, and 100 mM ammonium acetate (J.T.Baker) pH 9.5. Samples were made by the sequential addition of 800 µL of solution or buffer, 150 µL acetonitrile (MeCN), and 50 µL of saturated DnsCl (~57mM stock) dissolved in MeCN into a microcentrifuge tube. Immediately after the addition of DnsCl, the sample was vortexed and transferred to a quartz cuvette. Spectra (280–750 nm) were collected every 30 s for 5 min, every one minute from 5 to 16 min, and at 20 min and were baseline corrected at 650 nm. Six replicates were completed for each solution or buffer tested.

### 4.2. Protein Analysis with Native Mass Spectrometry and Ion Mobility Mass Spectrometry

Horse myoglobin (Sigma, product #M1882) and yeast alcohol dehydrogenase (Sigma, product #A3263) samples were prepared in 100 mM ammonium acetate at pH 5, 6, 7, 8.2, and 9.5 with and without organic solvent (2% and 20% MeCN). All samples were introduced with in-house prepared gold-coated borosilicate glass capillaries [33] and analyzed in positive mode on a Synapt G2-*Si* electrospray time-of-flight instrument (Waters, Milford, MA, USA). General operating parameters were source temperature of 60 °C, helium cell gas flow of 90 mL/min, IMS gas flow of 60 mL/min, trap DC bias of 60 V, trap wave velocity of 300 m/s, trap wave height of 7.1 V, IMS wave velocity of 855 m/s, IMS wave height of 40.0 V, transfer wave velocity of 120 m/s, and transfer wave height of 5.0 V. Trap energy was increased stepwise from 10 to 100 V during acquisition. NMS and IMS parameters for ADH samples were the same as myoglobin except for helium cell gas flow of 0–120 mL/min, IMS gas flow of 40 mL/min, trap DC bias of 90 V, trap wave velocity of 900 m/s, trap wave height of 7.0 V, IMS wave velocity of 900 m/s, and transfer wave velocity of 64 m/s.

### 4.3. Evaluation of Labeling Conditions

Temporal labeling reactions were performed on 5 µM horse myoglobin in 100 mM sodium phosphate pH 6 and pH 7 and 100 mM sodium carbonate-bicarbonate pH 9.5. The reactions were initiated by the addition of saturated DnsCl dissolved in MeCN for a final organic solvent concentration of 2%. Reactions were quenched at 5, 15, and 30 min with the addition of 10% reaction volume of 1M ammonium acetate. Intact protein analysis was then performed on an Agilent 1290 ultrahigh pressure series chromatography stack (Agilent Technologies, Santa Clara, CA, USA) coupled directly to a Micro-TOF electrospray time-of-flight mass spectrometer (Bruker Daltonics, Billerica, MA, USA). Before infusion to the ionization source, samples were separated on an Onyx monolithic C18 column (1.0 × 100 mm, Phenomenex, Torrance, CA, USA) at 50 °C using a flow rate of 200 μL/min and the following program: 1.0 min, 10% B; 1.0–6.0 min, 10–70% B; 6.0–7.0 min, 70–90% B; 7.0–8.0 min, 90% B; 8.0–8.1 min, 90–10% B; 8.1–10.0 min, 10% B; where solvent A = 0.1% FA in water and solvent B = 0.1% FA in MeCN. Electrospray conditions were as follows: nebulizer 3.0 bar, drying gas at a flow rate of 7.0 L/min, drying temperature at 200 °C, capillary voltage at 4.5 kV, and capillary exit voltage at 150 V. Data were collected in positive mode at 1 Hz over the 200–3000 *m*/*z* scan range. Data processing and analysis were performed using Bruker Data Analysis v.4.2 (Bruker Daltonics). Charge deconvolution was performed using a maximum entropy algorithm for H^+^ adducts only and 0.1 *m*/*z* data point spacing. The low mass end was defined by the mass of the lightest component while the high mass end was defined as 3.3× of the heaviest component within the sample.

### 4.4. Stability of Model Proteins After Label Incorporation

Myoglobin and ADH were labeled with DnsCl in 100 mM ammonium acetate at pH 7 and 9.5 for 5, 15, and 30 min. The reaction was initiated with the addition of saturated DnsCl solution in MeCN at a concentration of 2% (*v*/*v*) and the reaction was quenched with the addition of excess ammonium acetate (provided as 10% reaction volume of 1M ammonium acetate). In addition, myoglobin at pH 6 was labeled in a phosphate buffer, quenched with the addition of 10% reaction volume of 1M ammonium acetate, and subsequently dialyzed against 100 mM ammonium acetate, pH 7 using a 3.5 kDa MWCO dialysis membrane (Thermo-Fisher). The structural stability of the labeled protein was then evaluated with NMS and IMS as described above in “Section 4.2
*Protein analysis with native mass spectrometry and ion mobility mass spectrometry*”. The collision cross-section (CCS) values of myoglobin samples were calculated using a calibration curve. The calibration curve was generated using experimentally measured drift times for 5 µM IMS standards of cytochrome C (Sigma), lysozyme (Sigma), and α-chymotrypsinogen A (Sigma) in 200 mM ammonium acetate and CCS_He_ values retrieved from the Bush CCS database [34,35].

## Figures and Tables

**Figure 1 ijms-26-00456-f001:**
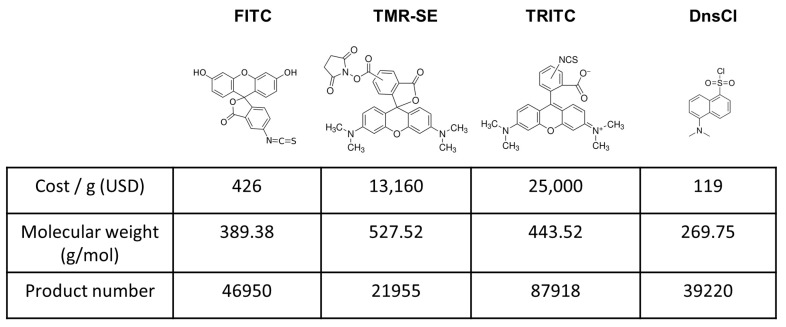
Dansyl chloride and commercial fluorescent probes for labeling lysine residues. The structure, cost, molecular weight, and product number of fluorescein 5(6)-isothiocyanate (FITC), 5(6)-Carboxytetramethylrhodamine N-succinimidyl ester (TMR-SE), tetramethylrhodamine isothiocyanate (TRITC), and dansyl chloride (DnsCl) are shown. The cost was determined using prices from Millipore Sigma obtained on 29 October 2024.

**Figure 2 ijms-26-00456-f002:**
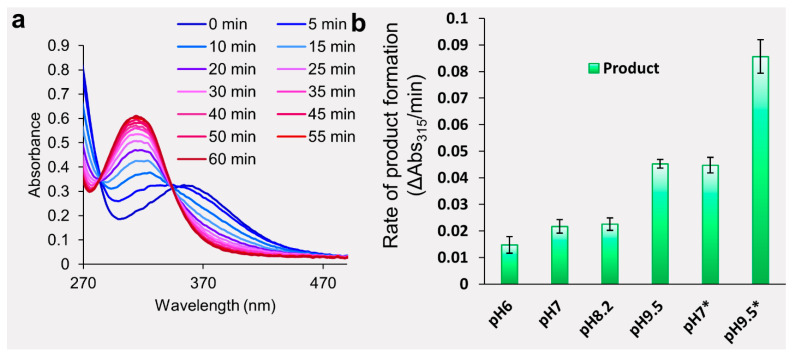
Stability of Dansyl in aqueous environments. (**a**) Hydrolysis of DnsCl at pH 7 in sodium phosphate buffer visualized by UV-VIS spectroscopy. Measurements are colored and transition from blue to red as the reaction proceeds. (**b**) The rate of the dansyl acid product formation in different buffers and solutions as measured by absorbance increase at 315 nm. From left to right: sodium phosphate buffers pH 6 and 7, HEPES buffer pH 8.2, carbonate-bicarbonate buffer pH 9.5, and ammonium acetate (indicated with *) pH 7 and 9.5. Values are the average of six replicates and error bars are ±1 standard deviation.

**Figure 3 ijms-26-00456-f003:**
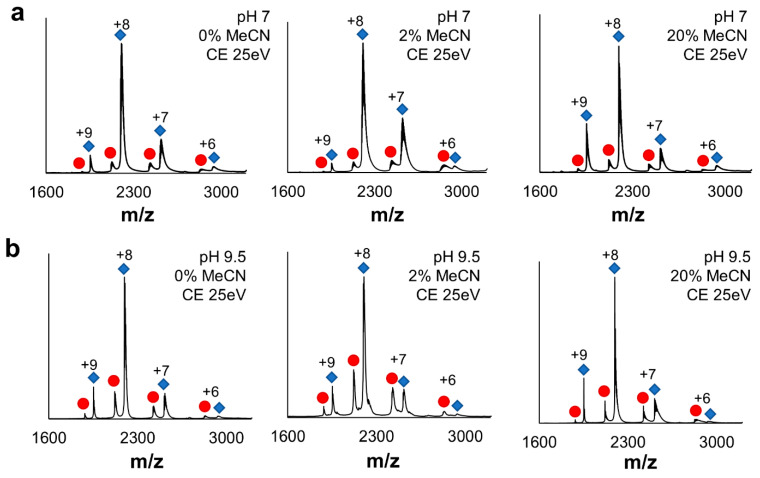
Characterization of myoglobin structural stability. Native mass spectra of myoglobin at (**a**) pH 7 and (**b**) pH 9.5 with 0, 2, and 20% acetonitrile (MeCN). Charge states are indicated above peaks. For each charge state, apo-myoglobin is on the left (red circle) and holo-myoglobin is on the right (blue diamonds). CE is collision energy.

**Figure 4 ijms-26-00456-f004:**
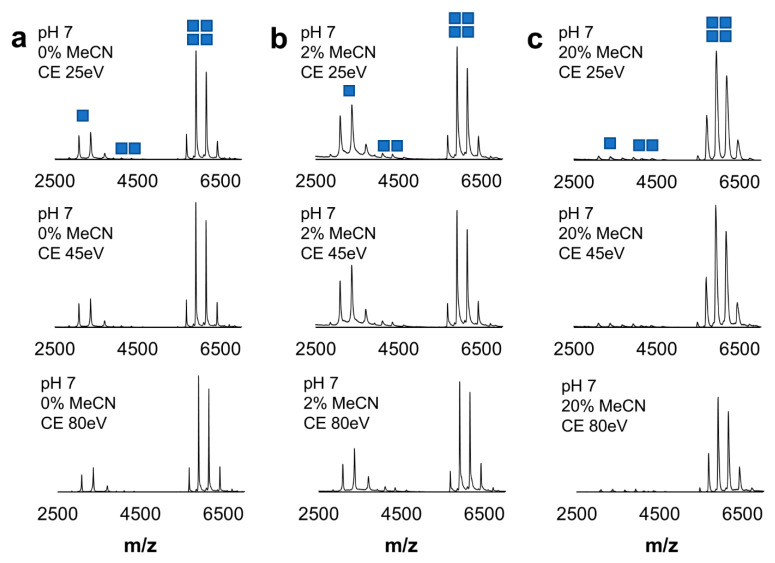
Characterization of ADH structural stability at pH 7. The structural integrity of alcohol dehydrogenase as an effect of organic solvent was investigated with native mass spectrometry. Native mass spectra are shown at increasing trap collision energy for ADH at pH 7 in (**a**) 0% MeCN, (**b**) 2% MeCN, and (**c**) 20% MeCN. Monomer, dimer, and tetramer charge envelopes are depicted in top spectra as one, two, and four squares, respectively.

**Figure 5 ijms-26-00456-f005:**
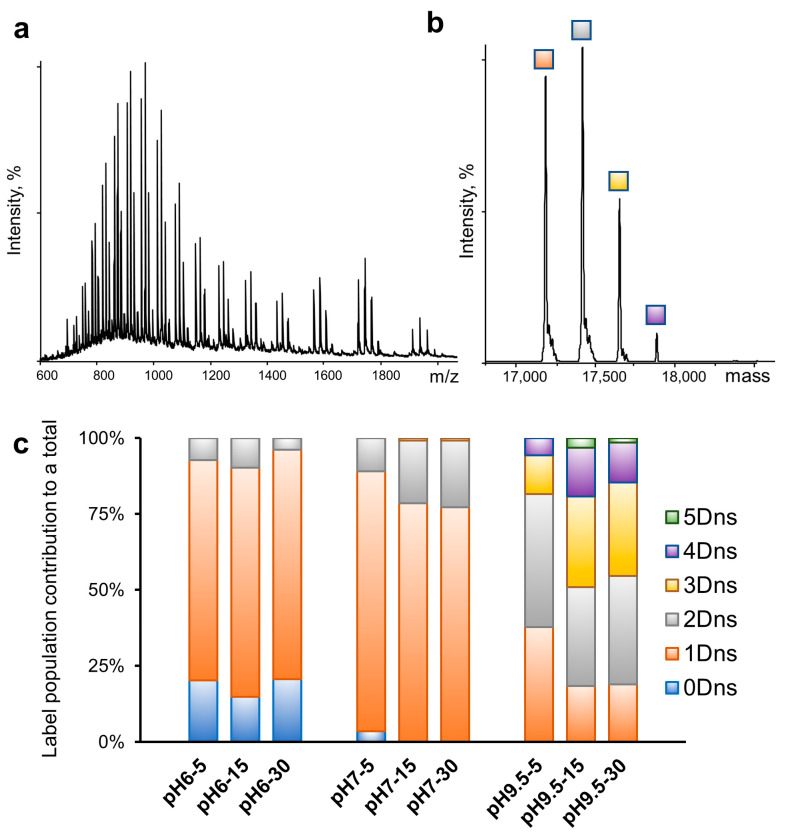
Robustness of protein dansylation reaction. (**a**) LC-MS of labeled myoglobin at pH 9.5 (carbonate-bicarbonate buffer) in 5 min; (**b**) deconvoluted spectrum of labeled myoglobin at pH 9.5 (carbonate-bicarbonate buffer); after 5 min exposure of the myoglobin to dansyl chloride the several populations of labeled protein were detected; major protein species contained from one to four dansyl labels incorporated (1 Dns; 17,184 Da, 2 Dns; 17,417 Da, 3 Dns; 17,651 Da, 4 Dns; 17,884 Da) (**c**) distribution of labeled myoglobin species after exposure to dansyl chloride for 5 min, 15 min, and 30 min at pH 9.5 (carbonate-bicarbonate), pH 7 (sodium phosphate buffer), and pH 6 (sodium phosphate buffer).

**Figure 6 ijms-26-00456-f006:**
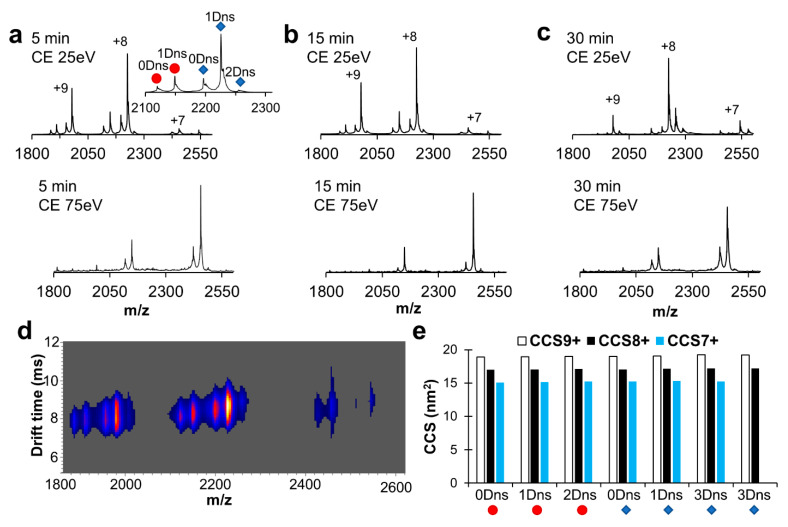
Effect of label load on structure stability for myoglobin. Native mass spectra are shown for myoglobin labeled with dansyl chloride in pH 7 ammonium acetate with 2% MeCN for (**a**) 5 min, (**b**) 15 min, and (**c**) 30 min with increasing trap collision energy. Inset in the first spectra of panel is of charge state +8 of myoglobin labeled for 5 min with collision energy of 25 eV. Apo-myoglobin is depicted as a red circle, and holo-myoglobin is depicted as blue diamonds. (**d**) Ion mobility spectra (mobilogram) of myoglobin labeled for 5 min at pH 7, collected with collision energy of 25 eV. Mobilogram intensity increases from blue to red. (**e**) CCS values of labeled myoglobin species at pH 7 after 5 min of labeling at a trap energy of 25 kV.

**Figure 7 ijms-26-00456-f007:**
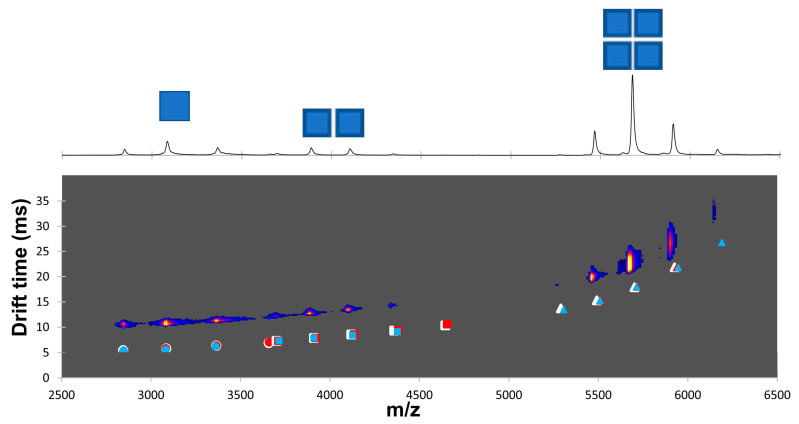
Effect of labeling on structure of ADH. Native mass spectra of unlabeled ADH at pH 7, in 0% MeCN (**above**). Monomer, dimer, and tetramer charge envelopes are depicted as one, two, and four squares, respectively. Drift times of labeled ADH are plotted on top of the unlabeled ADH mobilogram (**below**). ADH was labeled for 5 min (white), 15 min (red), and 30 min (blue). Note the mobilogram is offset by +5 msec from the labeled ADH data to allow the mobilogram intensity to be visible (intensity increases from blue to red). The monomeric, dimeric, and tetrameric data are plotted as circles, squares, and triangles, respectively.

## Data Availability

All the data supporting this article have been included in the main text and the Appendix A.

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
