# Peer review of "The Use of Dansyl Chloride to Probe Protein Structure and Dynamics"

_ijms, 2025, doi:10.3390/ijms26020456_

Round 1

Reviewer 1 Report

Comments and Suggestions for Authors

The paper by Larson et al demonstrates the possibility to use Dansyl labeling of proteins without significantly compromising fold state and structure. Dansyl, being a very small fluorescent probe, may enable multiple benefits, over much bulkier fluorescent probes. I liked the idea and the way the authors characterized the procedure they developed. However, at this point, the study is immature, and does not offer any application where dansylation can be used for protein characterization. The paper can be considered for publication after showing some examples. Without this, the overall significance of the study is low.

Major technical remarks

How can holo myoglobin be detected by LCMS, after denaturation on a C18 column and elution by ACN? By definition, this is Reversed Phase separation, which includes full denaturation, as the MS spectrum indeed shows.

Figure 6e - I find it very hard to believe that the holo and apo Myoglobin possess exactly the same CCS, as it appears from Fig 6e. I also find it hard to believe that dansylations do not affect CCS.

Additional remarks

Measured masses for all species in all spectra need to be shown.

To the best of my understanding, ammonia acetate and ammonium acetate are the same compound. I suggest to stick to one term, the more commonly used ammonium acetate.

Legend of Figure 2 - Please add to the legend the name/definition of the product (dansyl acid).

Legend of Figure 2 – What is CBC?

Legend of Figure 2 – Please indicate the meaning of the asterisks in 2B (ammonium acetate solutions).

All peaks in all figures (including supplementaries) need to be labeled with squares and circles, similar to the labeling in Figure 3.

Legend of Figure 3 - I think that the figure shows red circles and blue squares.

In what buffer was the labeling of Myoglobin performed, for the LC/MS measurements?

Figure 5b - The X axis should be mass, and not m/z.

Figure 6b-c - Zoom in of one charge state should be shown for all conditions. 

Figure 6e - 

Mobilogram in Figure 7 is not clear. Better keep coloring of the background similar to Figure 6.

Figure 7 – why was CCS not measured for labeled and unlabeled ADH?

Reviewer 2 Report

Comments and Suggestions for Authors

The manuscript presents a comprehensive study on Dansyl chloride as a chemical probe for investigating protein structure and dynamics using two model protein systems: myoglobin and alcohol dehydrogenase (ADH). The authors examined the protein stability in acetonitrile (up to 20%) and pH 6-9.5. Overall, the manuscript is well-written, and the experiments were designed systematically.

However, the following are the suggestions for improving this work. 

1) The authors need to add the functional assays for two proteins after adding the dansyl chloride to the protein (wild-type and modified by dansyl chloride). Although there might be no change in protein stability in acetonitrile (up to 20%), the authors need to address whether the protein's function may be changed or unchanged by modification. 

2) Using the dialysis or buffer exchange method, the authors may remove the organic solvent from the protein sample solution after completing the chemical reaction with dansyl chloride. However, the high concentration of acetonitrile solvent present in the sample may cause a protein unfolding. The authors may perform the buffer exchange (or dialysis) experiment to see whether protein function is recovered. 

Round 2

Reviewer 1 Report

Comments and Suggestions for Authors

I have no major remarks, other than on the calculated masses. A list peak m/z values is not sufficient. A list of measured masses for all spectra is mandatory.

Author Response

Measured masses for all spectra have been incorporated into the supplemental data tables.

Reviewer 2 Report

Comments and Suggestions for Authors

Although the authors may perform the protein functional assays later, the current manuscript is enough to be published in IJMS in the current form. I would like to recommend this manuscript to be accepted. 

Author Response

Thank you for reviewing our manuscript and recommending it for publication.